# *DACH1* mutation frequency in endometrial cancer is associated with high tumor mutation burden

**McKayla J. Riggs**[1]☯, **Nan Lin**[2]☯, **Chi Wang**[3,4‡], **Dava W. Piecoro**[5‡], **Rachel W. Miller**[1,4‡], **Oliver A. Hampton**[6‡], **Mahadev Rao**[7‡], **Frederick R. Ueland**[1,4‡], **Jill M. Kolesar**[1,2,4]☯*

**1** Division of Gynecologic Oncology, Department of Obstetrics and Gynecology, University of Kentucky, Lexington, Kentucky, United States of America, **2** College of Pharmacy, University of Kentucky, Lexington, Kentucky, United States of America, **3** Department of Biostatistics, College of Public Health, University of Kentucky, Lexington, Kentucky, United States of America, **4** Markey Cancer Center, University of Kentucky, Lexington, Kentucky, United States of America, **5** Division of Pathology, Department of Pathology and Laboratory Medicine, University of Kentucky, Lexington, Kentucky, United States of America, **6** Department of Bioinformatics and Biostatistics, M2Gen, Tampa, Florida, United States of America, **7** Department of Pharmacy Practice, Manipal College of Pharmaceutical Sciences, Manipal Academy of Higher Education, Manipal, Karnataka, India

☯ These authors contributed equally to this work.
‡ These authors also contributed equally to this work.
* jill.kolesar@uky.edu

**Data Availability Statement:** Authors received no special privileges in accessing the data. Raw data cannot be shared because they are both potentially identifying and contain sensitive patient data,

## Abstract

### Objective

*DACH1* is a transcriptional repressor and tumor suppressor gene frequently mutated in melanoma, bladder, and prostate cancer. Loss of *DACH1* expression is associated with poor prognostic features and reduced overall survival in uterine cancer. In this study, we utilized the Oncology Research Information Exchange Network (ORIEN) Avatar database to determine the frequency of *DACH1* mutations in patients with endometrial cancer in our Kentucky population.

### Methods

We obtained clinical and genomic data for 65 patients with endometrial cancer from the Markey Cancer Center (MCC). We examined the clinical attributes of the cancers by *DACH1* status by comparing whole-exome sequencing (WES), RNA Sequencing (RNASeq), microsatellite instability (MSI), and tumor mutational burden (TMB).

### Results

Kentucky women with endometrial cancer had an increased frequency of *DACH1* mutations (12/65 patients, 18.5%) compared to The Cancer Genome Atlas (TCGA) endometrial cancer population (25/586 patients, 3.8%) with p-value = 1.04E-05. *DACH1* mutations were associated with increased tumor mutation count in both TCGA (median 65 vs. 8972, p-value = 7.35E-09) and our Kentucky population (490 vs. 2160, p-value = 6.0E-04). *DACH1* mutated patients have a higher tumor mutation burden compared to *DACH1* wild-type (24

including geographic location, dates of diagnosis and dates of testing and receiving a medication. In addition, there are contractual agreements between the University of Kentucky and the Kentucky Cancer Registry precluding data sharing. Any requests for data must be submitted to: Jacyln K. McDowell, Epidemiologist, Kentucky Cancer Registry 2365 Harrodsburg Rd, Suite A230 Lexington, KY 40504 859-218-2228 Jaclyn. mcdowell@uky.edu.

**Funding:** Funding for this project was received by the NCI Cancer Center Support Grant (P30 CA177558). Dr. Piecoro's spouse's employment by Exelixis, Inc, is noted but completely unrelated to this project. His funding organization did not play a role in the study design, data collection and analysis, decision to publish, or preparation of the manuscript in any way. Dr. Hampton is employed by M2Gen and serves as the chief officer of bioinformatics over the ORIEN pipeline. M2Gen did not play a role in the study design, data collection and analysis, or decision to publish. He and his department assisted in separating our patients into DACH1 mutated vs wild-type cohorts through the ORIEN software and provided the information regarding the ORIEN genomic processing pipeline for this paper. Dr. Hampton was not involved in the subsequent bioinformatics analysis. He did provide final proofreading of the manuscript prior to submission. The specific roles of these authors are articulated in the 'author contributions' section.

**Competing interests:** The employment of Dr. Oliver Hampton by M2GEN does not alter our adherence to PLOS ONE policies on sharing data and materials.

vs. 6.02, p-value = 4.29E-05). *DACH1* mutations showed significant gene co-occurrence patterns with *POLE*, *MLH1*, and *PMS2*. *DACH1* mutations were not associated with an increase in microsatellite instability at MCC (MSI-H) (p-value = 0.1342).

## Conclusions

*DACH1* mutations are prevalent in Kentucky patients with endometrial cancer. These mutations are associated with high tumor mutational burden and co-occur with genome destabilizing gene mutations. These findings suggest *DACH1* may be a candidate biomarker for future trials with immunotherapy, particularly in endometrial cancers.

## Introduction

Uterine cancer is increasing in incidence and mortality in the United States. In 2020, an estimated 65,620 women will be diagnosed with endometrial cancer, making it the fourth most common female cancer, with an estimated 12,590 deaths [1]. Kentucky is an above-average risk region, with 29 new cases per 100,000 women compared to 27.6 per 100,000 women nationally [2]. The mean five-year survival rate for endometrial cancer is 81.2%, with more than 67% of patients diagnosed at an early stage. The survival rate decreases to 69% for locally metastatic and 16% for widely metastatic disease [3]. The treatment paradigm for endometrial cancer has been unchanged for some time. Current first-line therapy includes a combination of surgery, carboplatin and paclitaxel chemotherapy, and radiation depending on the stage and risk.

To molecularly categorize endometrial cancers, Kandoth and colleagues performed an integrated genomic, transcriptomic, and proteomic analysis of 373 endometrial carcinomas. They were able to classify these cancers into *POLE* ultramutated, microsatellite instability hypermutated (MSI-H), copy number low, and copy number high [4]. Uterine serous cancers and approximately 25% of high-grade endometrioid tumors were in the copy number high group and had frequent *TP53* mutations and poor prognosis. The majority of endometrioid cancers were in the copy number low group and were *TP53* wild-type with *PTEN* and *PIK3CA* commonly mutated. This group included low-grade endometrioid (60%), high-grade endometrioid endometrial cancers (8.7%), serous carcinomas (2.3%), and mixed-histology carcinomas (25%). The MSI-H group accounted for 28.6% of low-grade and 54.3% of high-grade endometrioid cancers studied. Frequently co-mutated genes included *PTEN* and *PIK3CA*. The *POLE* ultramutated groups accounted for 6.4% of low-grade and 17.4% of high-grade cancers and had improved progression-free survival. In addition, *POLE*-mutant microsatellite stable (MSS) tumors have been associated with high tumor mutation burden (TMB) in endometrial cancer [4].

The *Drosophila dachshund* (*dac*) gene (*DACH1*) was initially identified as critical to *Drosophila* eye development and is an essential member of the retinal determination gene network, responsible for normal organogenesis [5]. *DACH1* is a known tumor suppressor gene in breast, colon, and renal cancer and frequently mutated in melanoma, bladder, and prostate cancer. Most well-studied in breast carcinoma, *DACH1* is expressed in normal mammary epithelium with significantly reduced expression found in mastopathy, ductal carcinoma, and lobular carcinoma *in situ* [6]. *DACH1* expression was reduced or lost in invasive breast cancer patients with a poor prognosis [7], with its expression inversely related to tumor diameter, stage, and nodal metastasis, and directly associated with increased survival time [6]. While less

studied in uterine cancer, nearly all normal endometrial samples show nuclear expression of *DACH1*, with *DACH1* expression lost in more than half of endometrial cancers. Loss of *DACH1* expression is associated with poor prognostic factors, including higher FIGO surgical stage, positive peritoneal cytology, and lymph node positivity in endometrial cancer [8].

*DACH1* is a transcriptional co-repressor that functions as part of a DNA binding complex and regulates gene transcription. DACH1 is also an endogenous regulator of cyclin D1, with loss of *DACH1* resulting in increased cyclin D1, which is required for the G1/S transition. Nuclear expression of cyclin D1 is rarely observed in healthy endometrial tissue, while the majority of uterine cancers express cyclin D1, and cyclin D1 expression predicts poor survival [8, 9].

Our primary objective was to determine the frequency of *DACH1* mutations in our population and their association with other tumor suppressor genes, tumor mutation burden, microsatellite instability, and clinical risk factors.

## Methods

### Study design

ORIEN is a cancer precision medicine initiative initially developed by the Moffitt Cancer Center [10, 11]. It has evolved into a consortium research network of nineteen U.S. cancer centers, including the MCC, the only NCI designated cancer center in Kentucky, who joined the alliance in December 2017. All ORIEN alliance members utilize a standard protocol: Total Cancer Care (TCC)®. TCC is a prospective cohort study with whole-exome tumor sequencing, RNA sequencing, germline sequencing, and lifetime follow up. Nationally, over 250,000 participants have enrolled. As part of this study, participants agree to have their clinical data followed over time, to undergo germline and tumor somatic sequencing, and to be contacted in the future if an appropriate clinical trial becomes available [12]. At MCC, a buccal swab is used at enrollment for germline testing.

The Kentucky Cancer Registry (KCR) is a population-based central cancer registry for the Commonwealth of Kentucky. All cases of cancer diagnosed and/or treated in Kentucky are required to be reported to the KCR by state statute (KRS 214.556). Data elements reported to the registry consist of demographic and clinical information including genetic data. The final data set consolidated the linked demographic data with the genomic data from the enrolled patients pulled for the study. The Cancer Research Informatics Shared Resource Facility (CRI SRF) served as the honest broker and assisted with the distribution of clinical and genetic data stored in the KCR. A contractual agreement was previously established through M2GEN ORIEN/Total Cancer Care and the Kentucky Cancer Registry to allow data sharing. At the time of data receipt, all data was fully anonymized prior to analysis and the authors did not receive any special privileges in accessing the data.

### Study population

Patients presenting to Markey Cancer Center between December 1, 2018, and May 31, 2019, were invited to enroll in the parent trial, Total Cancer Care prospective cohort study. The study was offered to all eligible patients, and subjects were recruited during routine clinic visits. Treating physicians informed the patient about the study, and designated study coordinators assisted with enrollment and the formal consent process. Eligible patients were 18 years of age or older and had a diagnosis of cancer. A total of 65 patients with endometrial cancer enrolled at MCC were included in the analysis. To be included, each patient had to be ≥18 years of age, enrolled in TCC, and have both somatic and germline tumor whole exome sequencing results available. Patients were assigned a TCC ID number and otherwise de-

identified by the Kentucky Cancer Registry prior to analysis. The TCC ID number allowed the linkage of clinical data with genomic data through the CRI SRF honest broker. The study was conducted in accordance with the U.S. Common Rule, approved by the University of Kentucky Institutional Review Board (IRB #50767), and the investigators had obtained informed written consent from all subjects enrolling in TCC. Demographic variables, such as age at diagnosis, body mass index, race, and geographic location, were extracted from the linked KCR data and included in the analysis. Age at diagnosis was dichotomized to less than or equal to 64 and 65 and older. Clinical variables included cancer type, AJCC stage at diagnosis, and clinical comorbidities. Based on the frequency of cancer cases, types of cancer at diagnosis were grouped into several broader categories. Cancer stages were dichotomized to early (stage I-II) and late-stage (stage III-IV). The county of current residence and patient zip code were used to define Appalachia status as Appalachian or non-Appalachian. RNA sequencing was available for 52 patients. Tumor mutation burden and microsatellite instability data were available for 55 patients. The TCGA PanCancer Atlas dataset was used for comparison through cBioPortal.org, utilizing endometrial cancer and carcinosarcoma subgroups for analysis totaling 586 patients, which can be accessed here: https://www.cbioportal.org/results/ cancerTypesSummary?cancer_study_list=ucec_tcga_pan_can_atlas_2018%2Cucs_tcga_pan_ can_atlas_2018&Z_SCORE_THRESHOLD=2.0&RPPA_SCORE_THRESHOLD=2.0&data_ priority=0&profileFilter=0&case_set_id=all&gene_list=DACH1&geneset_list=%20&tab_ index=tab_visualize&Action=Submit.

## Sequencing methods (RUO)

ORIEN Avatar specimens undergo DNA and RNA extraction. For frozen and optimal cutting temperature (OCT) tissue DNA extraction, Qiagen QIASymphony DNA purification is performed, generating 213 bp average insert size. For frozen and OCT tissue RNA extraction, Qiagen RNAeasy plus mini kit is performed, generating 216 base pair (bp) average insert size. For formalin-fixed paraffin-embedded (FFPE) tissue, Covaris Ultrasonication FFPE DNA/RNA kit is utilized to extract both DNA and RNA, generating 165 bp average insert size. Preparation of M2GEN Whole Exome Sequencing (WES) libraries involves hybrid capture using an enhanced Integrated DNA Technology (IDT) WES kit (38.7 Mb) with additional custom-designed probes for double coverage of 440 cancer genes. Library hybridization is performed at either single or 8-plex and sequenced on an Illumina NovaSeq 6000 instrument generating 100 bp paired reads. WES is performed on tumor/normal matched samples with the normal covered at 100X and the tumor covered at 300X (additional 440 cancer genes covered at 600X) depth. We performed both tumor/normal concordance and gender identity quality control checks. The minimum threshold for hybrid selection is >80% of bases with >20X fold coverage; M2GEN WES libraries typically meet or exceed 90% of bases with >50X fold coverage for tumor and 90% of bases with >30X fold coverage for normal samples.M2GEN RNA sequencing (RNAseq) is performed using the Illumina TruSeq RNA Exome with single library hybridization, cDNA synthesis, library preparation, sequencing (100 bp paired reads at Hudson Alpha, 150 bp paired reads at Fulgent) to a coverage of 100M total reads / 50M paired reads.

## Bioinformatics

The bioinformatics pipeline was developed by M2Gen (Fig 1). The raw reads of WES and RNAseq data were saved in a fastq format. The adapter sequences were first trimmed by Bbduk software using paired-end read option. Reads were then mapped to the human genome using BWA-MEM with paired-end read option, which follows an alignment algorithm that aligns sequence reads or long query sequences against a large reference genome. GRCh38/

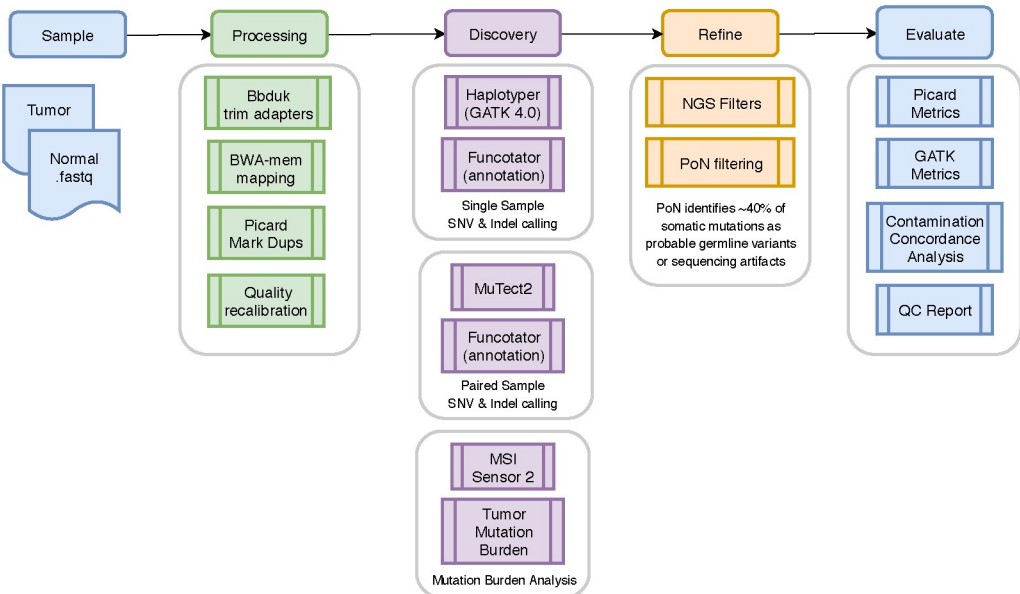

**Fig 1. Flow diagram illustrating the bioinformatics pipeline used by M2Gen.** Republished from M2Gen (https://www.m2gen.com/) under a CC BY license, with permission from Oliver A. Hampton, original copyright 2019.

hg38 human genome reference sequencing and GenCode build version 32 were used as the reference genome. We performed normalization, expression modeling, and difference testing using edgeR [13, 14]. The normalization of RNAseq counts was conducted with calcNormFactors function in edgeR, which normalizes the library sizes by finding a set of scaling factors to minimize the log-fold changes between the samples for most genes. The default method, a trimmed mean of M-values (TMM), was utilized to compute the scale factors between each pair of samples to provide the effective library size in the downstream RNAseq analysis. Next, the common dispersion was estimated from the housekeeping genes and libraries as a single group with function estimateDisp, which was controlled in differential analysis. Further, the expression model and differential expression analysis were completed using the functions glmFit, and statistics were controlled for multiple comparison using false discovery rate, which is defined as the expected proportion of false positives among all significant tests.

## Statistics and analysis

We performed a descriptive analysis of clinical variables and disease-related prognostic factors, including age, BMI, tumor grade, tumor stage, recurrence, tobacco usage, Appalachian status, and histology tumor subtypes. We compared categorical and continuous variables using the Chi-square test or Fisher's exact test and Student T-test, respectively. We performed a comparative analysis between ORIEN and TCGA datasets utilizing the Fisher's exact test. We calculated co-occurrence using the Fisher's exact test. Tumor mutation burden (TMB) and microsatellite instability (MSI) were calculated with the Wilcoxon rank sum test. Of the 65 patients, 55 had microsatellite instability and TMB data available. TMB was calculated using the count of non-synonymous somatic mutations (single nucleotide variants and small insertions/deletions, including missense, stop gain, stop loss and start loss mutations) per megacase in the coding region of the specific capture kit [15]. Percent of MSI was calculated using MSISensor2 (https://github.com/niu-lab/msisensor2, [16]) and dichotomized to MSI-H versus

MSS with a threshold of MSI-H $\geq$ 20% [17]. We used the Cox model for survival analyses [18], and corrected for dichotomized stage (high stage- III/IV, low stage- I/II), grade, and curves compared via the log-rank test. We performed all statistical analyses with R 3.6.3. The network analysis was performed using Qiagen's Ingenuity Pathway Analysis (IPA) system for core analysis of the RNA sequencing data and overlaid with the Global Molecular Network Overlay in the IPA knowledge base. Using IPA, canonical pathways, disease and functions, and gene networks were categorized based on differential gene expression.

## Results

Out of 65 patients, 12 had *DACH1* gene mutations as shown in Table 1. The mean age and BMI were 62 years and 36.4 kg/m$^2$, respectively. The majority of patients were stage I (47.7%), but a significant portion were stage III and IV (40%). Grade 3 disease (50.8%) was common, and 30.7% had grade 1 disease. Disease recurrence was present in 24.6% of the patients. Approximately 60% of the patients were from the Appalachian region, which mirrors the percentage of patients treated from the Appalachian region by MCC as a whole. The cell types were distributed as follows: 57% endometrioid, 23.1% high grade serous, 9.2% carcinosarcoma, and 7.7% mixed cell adenocarcinoma. One patient each had clear cell carcinoma and malignant mesonephroma. Twelve of the 65 patients had at least one deleterious mutation in the *DACH1* gene by whole exome sequencing. Eleven patients had point mutations, and six patients had more than a single mutation in the gene (range 1–7 point mutations). Two patients had a one-base-pair insertion, and one of these patients had a total of seven mutations with six point mutations and one base-pair-insertion (Fig 2 and Table 2).

There were no significant associations between *DACH1* mutation and clinical covariates, including grade, stage, or histology. Age is approaching significance with a p-value of 0.053, with *DACH1* mutations trending towards occurring more frequently in older patients, as shown in Table 3. Though not reaching statistical significance, 7/12 (58%) of the patients with *DACH1* mutations also had high-grade disease, compared to 26/53 (49%) of those who were wild-type. There was no statistical difference seen in Appalachian versus non-Appalachian patients nor the histologic subtype. *DACH1* gene mutations were not statistically associated with a microsatellite unstable genome in either the MCC cohort (p-value = 0.1342) or in the TCGA analysis through cBioPortal using the MSIsensor Score (p-value = 0.142) as shown in Table 4. Other commonly occurring driver mutations associated with microsatellite instability and genome instability were frequent in the *DACH1* patients.

We compared frequencies of commonly found driver mutations including *PTEN*, *PIK3CA*, *TP53*, *POLE*, and the Lynch Syndrome-associated genes (*MLH1*, *MSH2*, *MSH6*, *PMS2*) using the enrichment test to determine whether the frequency of gene mutations in the MCC cohort was similar to that of the TCGA PanCancer Atlas (PCA) endometrial carcinoma and carcinosarcoma datasets. We identified 3.8% (25/586 patients) *DACH1* gene mutations in uterine cancer somatic samples in the TCGA PCA of endometrial and carcinosarcoma patients, which was significantly lower than the 18.5% (12/65, p = 1.05E-05) seen in the MCC patient cohort. In addition, *MLH1* (22/65, 33.85%, p = 2.63E-13), *MSH2* (25/65, 38.46%, p = 2.87E-07*), MSH6* (28/65, 43.1%, p = 1.01E-11*), PMS2* (12/65, 18.5%, p = 3.79E-03*), and *POLE* (28/65, 43.08%, 5.39E-08) mutations were more common in the MCC cohort than the TCGA PCA, while mutation frequency in *PTEN*, *PIK3CA*, and *TP53* were not statistically difference in the two datasets. We performed a co-occurrence analysis using the Fisher's exact test to determine whether *DACH1* mutations are mutually exclusive or tend to co-occur with other gene mutations, with a significant co-occurrence pattern noted between *DACH1* and two of the four Lynch Syndrome associated genes, *MLH1* (p = 2.39E-04) and *PMS2* (p = 3.67E-07) as well as

**Table 1. Demographics of the Markey Cancer Center population.**

| Patient Demographics | |
|---|---|
| **Age** | 62.14 ± 10.79 |
| **BMI** | 36.38 ± 10.22 |
| **Race** | |
| Caucasian | 59 (90.8%) |
| African American | 5 (7.7%) |
| Asian | 1 (1.5%) |
| **Grade** | |
| 1 | 20 (30.7%) |
| 2 | 12 (18.5%) |
| 3 | 33 (50.8%) |
| **Clinical stage** | |
| I | 31 (47.7%) |
| II | 5 (7.7%) |
| III | 14 (21.5%) |
| IV | 12 (18.5%) |
| Unknown | 3 (4.6%) |
| **Clinical stage, early vs. late** | |
| Early stage (I-II) | 36 (55.4%) |
| Late stage (III-IV) | 26 (40.0%) |
| Unknown | 3 (4.6%) |
| **Tobacco use (smoking)** | |
| No | 43 (66.2%) |
| Yes | 21 (32.3%) |
| Unknown | 1 (1.5%) |
| **Documented recurrence** | |
| No | 42 (64.6%) |
| Yes | 16 (24.6%) |
| Unknown | 7 (10.8%) |
| **Appalachian status** | |
| Non-Appalachian | 25 (38.5%) |
| Appalachian | 40 (61.5%) |
| **Histologic subtype** | |
| Endometrioid | 37 (57.0%) |
| Mixed cell adenocarcinoma | 5 (7.7%) |
| Carcinosarcoma | 6 (9.2%) |
| Serous | 15 (23.1%) |
| Other: Clear cell, malignant mesonephroma | 2 (3.1%) |
| **DACH1** | |
| Mutated | 12 (18.5%) |
| Wild-type | 53 (81.5%) |

*DACH1* and *POLE* (p = 5.78E-08), shown in Table 5. Neither *MSH2* (p = 0.0628) nor *MSH6* (p = 0.264) co-occurred with *DACH1* in the MCC cohort, although this may be related to small sample size. The co-occurrence of *DACH1* with *MLH1*, *PMS2*, and *POLE* were replicated in the TCGA PCA dataset adjusting for FDR using the Benjamini-Hockberg procedure through cBioPortal, with a significant co-occurrence pattern also found with *DACH1* and *MSH2*, and *DACH1* and *MSH6* (q-value <0.001).

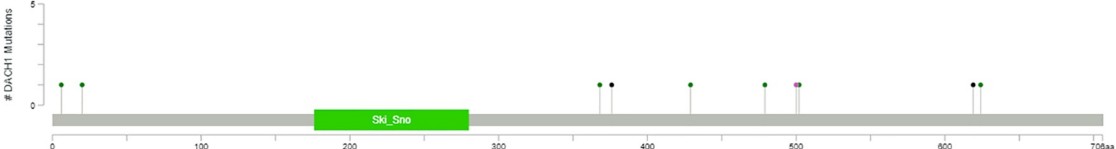

**Fig 2. Lollipop plot of *DACH1* gene mutations in the 65 patients in the Markey Cancer Center.** Missense mutations are green. Truncating mutations are black and include nonsense, nonstop, frameshift deletions, frameshift insertions, and splice site mutations. All other types of mutations are included as pink (excluding fusion and inframe deletion or insertions).

Given that *DACH1* plays a complex role in transcriptional repression, we performed a gene expression analysis using the RNA sequencing data in Qiagen's Ingenuity Pathway Analysis (IPA) to evaluate differences in expression and pathways to better understand the mechanism of action of *DACH1*. Of the 65 patients, 52 had RNA sequencing data available. A total of 2,599 genes were significantly differentially expressed (FDR values < 0.05) between the *DACH1* mutated patients and wild-type (Fig 3), with a large proportion of these being upregulated in the setting of mutated *DACH1*. The top ten upregulated and downregulated differentially expressed genes comparing *DACH1* mutated patients to wild-type are displayed in Table 6a and 6b. In the top ten upregulated genes, many are involved in transcription regulation and

**Table 2. Description of *DACH1* gene mutations in the 12 *DACH1* mutated patients at Markey Cancer Center.**

| Patient | Protein Change | Mutation Type | Variant Type | Start Pos | End Pos | Ref | Var |
|---|---|---|---|---|---|---|---|
| 1 | | 3'UTR | SNP | 71439896 | 71439896 | A | C |
| 2 | | 3'UTR | SNP | 71439710 | 71439710 | C | A |
| 2 | A624T | Missense_Mutation | SNP | 71479169 | 71479169 | C | T |
| 2 | T429P | Missense_Mutation | SNP | 71572854 | 71572854 | T | G |
| 2 | A6T | Missense_Mutation | SNP | 71866754 | 71866754 | C | T |
| 3 | *479* | Intron | SNP | 71559804 | 71559804 | G | A |
| 4 | *672* | Intron | SNP | 71475641 | 71475641 | T | A |
| 4 | E619* | Nonsense_Mutation | SNP | 71479184 | 71479184 | C | A |
| 4 | *479* | Intron | SNP | 71557209 | 71557209 | A | T |
| 5 | *575* | Intron | SNP | 71479386 | 71479386 | A | T |
| 6 | | 3'UTR | SNP | 71438559 | 71438559 | G | T |
| 6 | N368K | Missense_Mutation | SNP | 71630578 | 71630578 | G | C |
| 6 | *322* | Intron | SNP | 71674937 | 71674937 | T | G |
| 7 | P502L | Missense_Mutation | SNP | 71557089 | 71557089 | G | A |
| 7 | P500 = | Silent | SNP | 71557094 | 71557094 | A | G |
| 8 | | 3'UTR | SNP | 71438493 | 71438493 | G | T |
| 8 | | 3'UTR | SNP | 71439241 | 71439241 | C | T |
| 8 | I20T | Missense_Mutation | SNP | 71866711 | 71866711 | A | G |
| 9 | | 3'UTR | SNP | 71438127 | 71438127 | C | T |
| 9 | | 3'UTR | SNP | 71438479 | 71438479 | A | C |
| 9 | | 3'UTR | SNP | 71438897 | 71438897 | C | T |
| 9 | *695* | Intron | SNP | 71464674 | 71464674 | G | T |
| 9 | P479L | Missense_Mutation | SNP | 71557158 | 71557158 | G | A |
| 9 | *433* | Intron | SNP | 71572766 | 71572766 | G | T |
| 9 | *376* | Intron | INS | 71573375 | 71573375 | T | TA |
| 10 | | 3'UTR | SNP | 71439315 | 71439315 | G | A |
| 11 | *575* | Intron | SNP | 71479371 | 71479371 | T | A |
| 12 | X376_splice | Splice_Region | INS | 71573016 | 71573016 | T | TA |

**Table 3. Covariate analysis of *DACH1* mutated patients compared to wild-type.**

| Covariates | DACH1 | | P-value[a] |
|---|---|---|---|
| | **WT N = 53** | **M N = 12** | |
| **Age** | 61.15 ± 11.25 | 66.50 ± 7.35 | **0.05267** |
| **BMI** | 36.91±10.65 | 34.03±8.03 | **0.3047** |
| **Race** | | | **0.3746** |
| Caucasian | 49 (75.4%) | 10 (15.4%) | |
| African American | 3 (4.6%) | 2 (3.1%) | |
| Asian | 1 (1.5%) | 0 (0.0%) | |
| **Grade** | | | **0.9121** |
| 1 | 17 (32%) | 3 (25%) | |
| 2 | 10 (18.8%) | 2 (16.7%) | |
| 3 | 26 (49%) | 7 (58.3%) | |
| **Clinical stage** | | | **0.4144** |
| I | 23 (43.4%) | 8 (66.7%) | |
| II | 5 (9.4%) | 0 (0.0%) | |
| III | 13 (24.5%) | 1 (8.3%) | |
| IV | 10 (18.9%) | 2 (16.7%) | |
| Unknown | 2 (3.8%) | 1 (8.3%) | |
| **Clinical stage, early vs. late** | | | **0.3316** |
| Early stage (I-II) | 28 (52.8%) | 8 (66.7%) | |
| Late stage (III-IV) | 23 (43.4%) | 3 (25%) | |
| Unknown | 2 (3.8%) | 1 (8.3%) | |
| **Tobacco use (smoking)** | | | **1** |
| No | 35 (66%) | 8 (66%) | |
| Yes | 17 (32%) | 4 (6.2%) | |
| Unknown | 1 (1.9%) | 0 (0%) | |
| **Documented recurrence** | | | **1** |
| No | 33 (50.8%) | 9 (13.8%) | |
| Yes | 13 (20.0%) | 3 (4.6%) | |
| Unknown | 7 (13.2%) | 0 (0%) | |
| **Appalachian status** | | | **1** |
| Non-Appalachian | 20 (30.8%) | 5 (7.7%) | |
| Appalachian | 33 (50.8%) | 7 (10.8%) | |
| **Histologic subtype[b]** | | | **0.9473** |
| Endometrioid | 31 (47.7%) | 6 (9.23%) | |
| Mixed cell adenocarcinoma | 4 (6.2%) | 1 (1.5%) | |
| Carcinosarcoma | 5 (7.7%) | 1 (1.5%) | |
| Serous | 12 (18.5%) | 3 (4.6%) | |

[a] P-values were calculated using the Fisher's exact test for categorical variables and using the student t-test for continuous variables.

[b] Single cases each of malignant mesonephroma (*DACH1* mutated) and clear cell carcinoma (*DACH1* wild-type) excluded from covariate analysis.

cell signaling. In contrast, the top downregulated genes were part of the immune system response and the transcription of ER/PR receptors. We performed an in-depth pathway analysis utilizing the RNA sequencing data to determine cell-specific pathways impacted by *DACH1* mutations, shown in Table 7 and Fig 4A and 4B. Of note, the most significant pathways involved were the breast cancer development pathway by a log value of 3.45, and catecholamine and transcriptional regulation pathways each by a log value of 3.19.

**Table 4. Genomic covariate analysis of *DACH1* mutated patients compared to wild-type at MCC (n = 65 patients).**

| Covariates | DACH1 | | Co-Occurrence P-value[a] |
|---|---|---|---|
| | WT N = 53 | M N = 12 | |
| PTEN (n = 40) | 32/40 | 8/40 | 0.94 |
| PIK3CA (n = 35) | 26/35 | 9/35 | 0.191 |
| TP53 (n = 31) | 23/31 | 8/31 | 0.255 |
| POLE Mutation (n = 28) | 17/28 | 11/28 | 5.78E-04 |
| MLH1 Mutation (n = 22) | 12/22 | 10/22 | 2.39E-04 |
| MSH2 Mutation (n = 25) | 17/25 | 8/25 | 0.046 |
| MSH6 Mutation (n = 28) | 19/28 | 9/28 | 0.264 |
| PMS2 Mutation (n = 12) | 8/12 | 4/12 | 3.67E-07 |
| Microsatellite Instability (n = 55) | | | 0.1342[b] |
| MSI-High (n = 7) | 4/7 | 3/7 | |
| Microsatellite Stable (n = 48) | 40/48 | 8/48 | |
| Tumor Mutation Burden | 6.02 | 24.0 | 4.29E-05[c] |

[a] P-values were calculated using the Fisher's exact test for categorical variables and using the student t-test for continuous variables. The co-occurrence analysis using Fisher's exact test was performed to determine whether *DACH1* mutations are mutually exclusive or tend to co-occur with other gene mutations.

[b] MSI data was only available for 11/12 DACH1 mutated patients and 44/53 of the DACH1 wild type patients.

[c] Tumor mutation burden p-value was calculated using the Wilcoxon rank sum test.

To better assess clinical applicability, we converted the pathway analysis to a heat map with analysis by disease and organ system (Fig 5A). The size of the box denotes the -log(p-value). The color of the boxes correlates with the z-score with the intensity of blue representing $z \leq 0$ and orange $z \geq 0$. Pathways related to cellular injury and cancer predominated, suggesting *DACH1* mutations lead to disease processes resulting in cellular injury and cancer (Fig 5A). Specific to cancer, several statistically significant p-values with z-score $\geq 0$ indicated overexpression, including pelvic cancer (-log[p-value] = 5.439, z-score = 0.391), genital cancer (-log[p-value] = 3.967, z-score = 0.391), and quantity of malignant tumor (-log[p-value] = 2.502, z-score = 1.254) (Fig 5B). *DACH1* was most associated with the cancer forming pathway followed closely by organismal injury and abnormalities, diseases of the endocrine system, and the gastrointestinal system (Fig 5C). Multiple pathways involved in cell cycle control, signaling, and development were also significantly differentially expressed between *DACH1* mutated and wild-type as assessed by RNA sequencing.

We performed network mapping using IPA with Global Network Overlay to determine the interplay of *DACH1* on genes found to be significantly altered between *DACH1* mutated patients and wild-type (Fig 6A). We note upregulated expression in red, with color intensity corresponding to increased significance. Downregulated expression is notated in green with color intensity again corresponding to increased significance. Network mapping results were filtered by statistically significant p-values with expression fold changes $\geq 0$ (Fig 6B). This revealed an interplay between three genes including *ASCL1* (3.070 expression fold change, p-value = 2.08E-03), *SOX2* (expression fold change 3.470, p-value = 1.84E-02), and *LHX1* (4.090 expression fold change, p-value = 1.58E-02) when comparing *DACH1* mutated patients to wild-type. Each of these three genes is involved in transcription regulation and cell cycle control. The top five network functions included differentiation of chromaffin cells (p-value = 3.97E-05), activation of DNA endogenous promoter (p-value = 6.05E-05), transcription of DNA (p-value = 2.64E-04), fusion of bone (p-value = 2.74E-04), and formation of the forebrain n (p-value = 5.0E-04).

**Table 5. Comparison of mutation frequency in endometrial cancer and carcinosarcoma between MCC and TCGA PanCancer Atlas (PCA).**

| | TCGA Frequency | MCC Frequency | P-value[a] |
|---|---|---|---|
| | | | MCC vs TCGA |
| **PTEN** | 62% | 62% | 1 |
| | 363/586 | 40/65 | |
| **PIK3CA** | 52% | 53.9% | 0.885 |
| | 305/586 | 35/65 | |
| **TP53** | 42% | 47.7% | 0.452 |
| | 246/586 | 31/65 | |
| **DACH1** | 3.8% | 18.5% | 1.05E-05 |
| | 25/586 | 12/65 | |
| **MLH1** | 6% | 33.9% | 2.63E-13 |
| | 35/586 | 22/65 | |
| **MSH2** | 8.9% | 38.5% | 2.87E-07 |
| | 52/586 | 25/65 | |
| **MSH6** | 11.1% | 43.1% | 1.01E-11 |
| | 65/586 | 28/65 | |
| **PMS2** | 7.2% | 18.5% | 3.79E-03 |
| | 42/586 | 12/65 | |
| **POLE** | 14.7% | 43.1% | 5.39E-08 |
| | 86/586 | 28/65 | |
| **MSI-H** | 14.5% | 12.7% | 0.719[b] |
| | 85/586 | 7/55 | |
| **MSS** | 85.5% | 87.3% | 0.719[c] |
| | 501/586 | 48/55 | |

[a] A comparison analysis between the MCC and TCGA PCA datasets was conducted utilizing the Fisher's exact test to determine significance.

[b] Fisher's exact test was used to compare MCC and TCGA PCA datasets for microsatellite status with no difference found.

Given *DACH1*'s control of transcription regulation and the cell cycle, we then evaluated its effect on tumor mutation burden and microsatellite instability. We first compared tumor mutation counts in the TCGA PCA endometrial cancer and carcinosarcoma cohorts between *DACH1* mutated patients and wild-type and found clinically significant differences with a median of 8972 in *DACH1* mutants vs. 65 in *DACH1* wild-type (p-value = 7.35e-09). We repeated this analysis in our MCC population with a median of 2160 in *DACH1* mutated patients vs. 490 in *DACH1* wild-type (p-value = 6E-04). We then compared TMB between *DACH1* wild-type and mutated patients in the MCC cohort. Of the 65 patients, 55 had microsatellite instability and TMB data available. As expected, given the marked difference in tumor mutation counts in both TCGA PCA and MCC, *DACH1* wild-type patients had a median TMB of 6.02 and *DACH1* mutated patients had a significantly higher median TMB of 24.0 (p-value = 4.29E-05) compared by the Wilcoxon rank sum test (Fig 7). Given the co-occurrence of mutations found in *DACH1* with *MLH1*, *POLE*, and *PMS2*, we then compared microsatellite instability between *DACH1* mutated patients and *DACH1* wild-type in the MCC cohort using the chi-square test, and no significance was found between the two groups (p-value = 0.2659) as shown in Table 8, with 3/12 *DACH1* mutated patients being MSI-H, and 8/12 being MSS, and one patient with MSI unavailable.

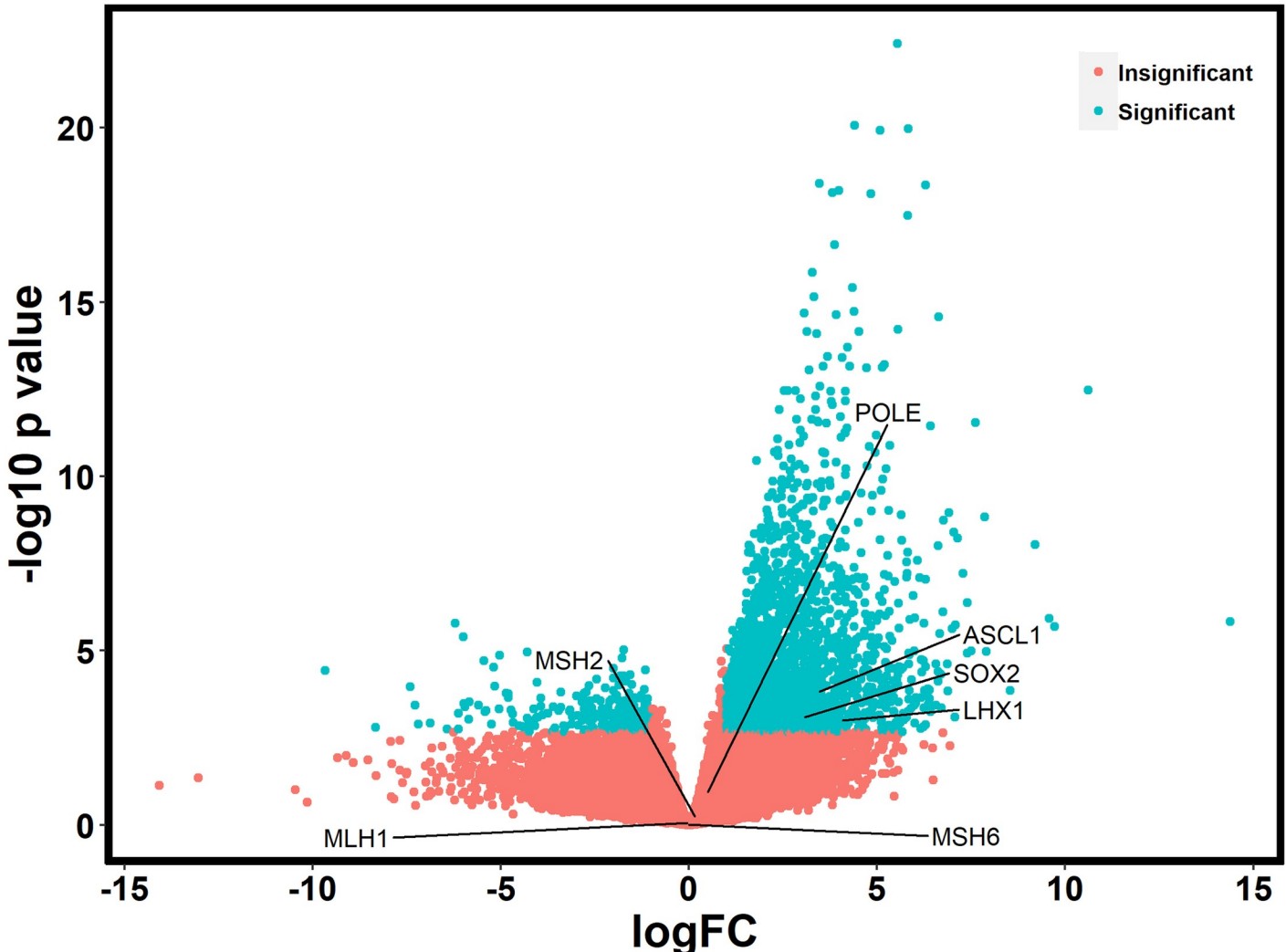

**Fig 3. Differential gene expression between *DACH1* mutated patients and wild-type.** A marked increase in significantly upregulated genes is noted.

We assessed overall survival analysis between MCC *DACH1* mutated patients and wild-type given prior studies suggesting an increase in stage, lymph node status, and metastasis with reduced *DACH1* expression using the Cox model, correcting for stage and grade. No significant difference was noted between *DACH1* wild-type and mutated patients (p-value = 0.803), with 80% of patients still alive at five years in both groups (Fig 8A). The result was similar in the TCGA PCA dataset with no difference in overall survival (p-value = 0.196) (Fig 8B). In TCGA PCA, at five years, 90.48% of the patients with the *DACH1* mutation were still alive, while 70.47% of patients who were *DACH1* wild-type group were still alive.

## Discussion

*DACH1* plays a critical role in cell cycle control and acts as a tumor suppressor gene in breast cancer [6]. Our network analysis further supports this role in endometrial cancer by revealing three essential upregulated genes and their pathways with significant differences in expression between *DACH1* mutated patients and wild-type, *ASCL1*, *SOX2*, *LHX1*. *ASCL1* and *SOX2* are

**Table 6. Top significant differentially expressed genes comparing *DACH1* mutated patients to wild-type and their associated pathways.** (A) Upregulated pathways in *DACH1* mutated patients compared to wild-type. (B) Downregulated pathways in *DACH1* mutated patients compared with wild-type.

| Upregulated | | | | |
|---|---|---|---|---|
| Genes | logFC | PValue | Qvalue (FDR) | Function |
| 1. F8A2 | 14.38482 | 1.50E-06 | 0.000206 | Vesicle trafficking |
| 2. CRH | 10.60565 | 3.31E-13 | 5.69E-10 | Cell-signaling |
| 3. HOXD12 | 7.861497 | 1.45E-09 | 6.52E-07 | Transcription regulation |
| 4. GFRA4 | 7.281125 | 6.03E-08 | 1.43E-05 | Growth factor signaling |
| 5. ELK2AP | 7.146813 | 6.04E-09 | 2.14E-06 | Transcription regulation |
| 6. CTAG2 | 6.992995 | 2.35E-06 | 0.00029 | Testis antigen |
| 7. DPP4 | 6.632547 | 2.66E-15 | 8.76E-12 | T-cell activation |
| 8. MAGEB1 | 6.616377 | 3.95E-05 | 0.00278 | Testis antigen |
| 9. EVX2 | 6.173953 | 1.83E-05 | 0.001525 | Transcription regulation |
| 10. PNMA5 | 5.811926 | 3.29E-18 | 1.95E-14 | Immune response |
| Downregulated | | | | |
| Genes | logFC | P-Value | Q-value (FDR) | Function |
| 1. CST4 | -9.66649 | 3.78E-05 | 0.002704 | Protease inhibitor |
| 2. PAGE1 | -8.33377 | 0.001611 | 0.040795 | Tumor antigen |
| 3. DEFA1 | -7.27866 | 0.000371 | 0.014545 | Immune System |
| 4. GP2 | -7.19004 | 0.0013 | 0.035124 | Immune System |
| 5. INSL4 | -6.87798 | 0.001215 | 0.033501 | Cell-signaling |
| 6. KRTAP4-4 | -6.82882 | 0.029918 | 0.25257 | Development |
| 7. MYBPC1 | -6.12666 | 0.001753 | 0.043191 | Cell Structure |
| 8. SRARP | -6.12316 | 0.00063 | 0.021442 | Transcription of E2/PR receptors |
| 9. DMBT1 | -6.00111 | 3.99E-06 | 0.000454 | Immune System |
| 10. LFT | -5.97798 | 0.000334 | 0.013478 | Immune System |

important transcription factors involved in cell cycle regulation via interaction with Cyclin D [19–21], and *LHX1* is a DNA-binding transcription factor. We anticipate that *DACH1* mutations result in loss of transcriptional repression of these regulators resulting in uncontrolled cell cycle progression in endometrial cancer, similar to *DACH1*'s control of the cell cycle via cyclin D1 in breast cancer [8, 22].

*POLE* is a tumor suppressor gene involved in nucleotide excision repair, which is mutated in 7–15% of endometrial cancers [19] and is associated with a good prognosis and a high

**Table 7. Pathway analysis of genes differentially expressed between *DACH1* mutated patients and wild-type.**

| Ingenuity Canonical Pathways | -log(p-value) |
|---|---|
| 1. Breast Cancer Regulation by STMN1 | 3.45 |
| 2. Catecholamine Biosynthesis | 3.19 |
| 3. Transcriptional Regulatory Network in Embryonic Stem Cells | 3.19 |
| 4. Serotonin and Melatonin Biosynthesis | 2.53 |
| 5. Methionine Salvage II (Mammalian) | 2.06 |
| 6. FXR/RXR Activation | 2.01 |
| 7. LPS/IL-1 Mediated Inhibition of RXR Function | 1.9 |
| 8. Stearate Biosynthesis I (Animals) | 1.86 |
| 9. Complement System | 1.83 |
| 10. Thyroid Hormone Metabolism II (via Conjugation and/or Degradation) | 1.77 |

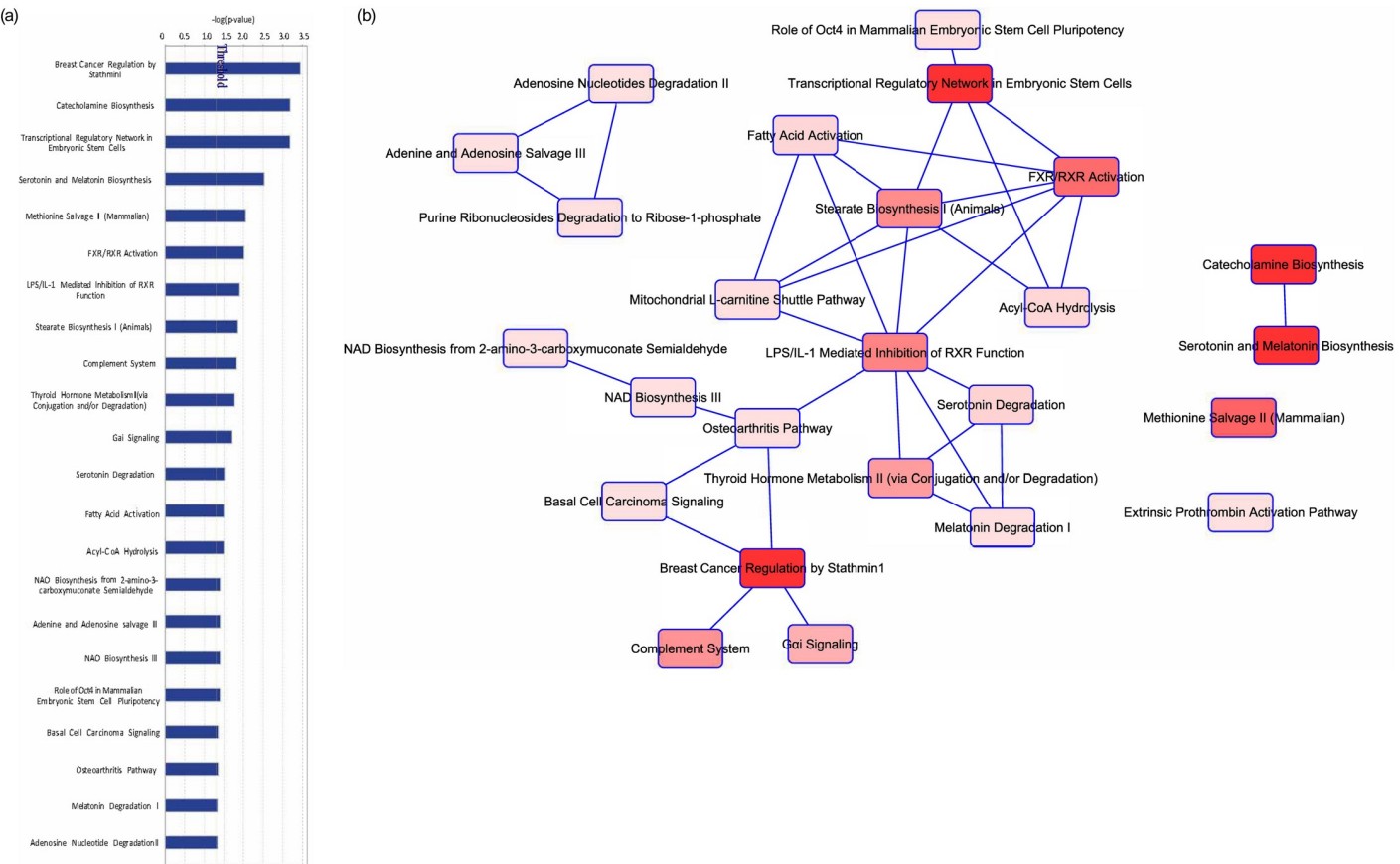

**Fig 4. Differential expression analysis of *DACH1* mutated versus wild-type patients.** (A) Pathway analysis of genes differentially expressed between *DACH1* mutated patients and wild-type. (B) Network analysis of the pathways differentially expressed between *DACH1* mutated patients and wild-type.

TMB. In our population, we identified a high frequency of both *DACH1* and *POLE* mutations when compared to the TCGA PCA, and that *POLE* and *DACH1* are significantly co-mutated. Notably, mutation frequencies of other common driver genes, *PTEN*, *PIK3CA*, and *TP53*, were consistent between the two datasets. In population studies, approximately 25% of endometrial tumors exhibit MSI-H status by IHC. Of these, the majority (~85%) are explained by hypermethylation of the MLH1 promoter, approximately 5% by germline mutations in Lynch-associated genes (*MLH1*, *MSH2*, *MSH6*, *PMS2*) and the remaining 10% by somatic mutations, unusual germline mutations not covered by clinical panels, or *POLE* mutations [23]. In our Kentucky population, mutation frequency in *MLH1*, *MSH2*, *MSH6*, and *PMS2* was approximately 34%, 38%, 43%, and 19%, respectively, all significantly higher than reported by TCGA and with significant co-occurrence between *DACH1* and *MLH1* and *PMS2*, despite similar rates seen in *TP53*, *PTEN*, and *PIK3CA*. Since co-occurrence of mutations typically occurs among functionally related genes that work together to promote tumorigenesis [24], we hypothesize that *DACH1* and DNA repair genes like *POLE* partner to halt the cell cycle and repair DNA and that concurrent mutation of these tumor suppressors drives oncogenesis in a subset of patients with endometrial cancer. We also suggest that this sub-type of endometrial cancer, while present in the TCGA PCA, is significantly overrepresented in Kentucky patients

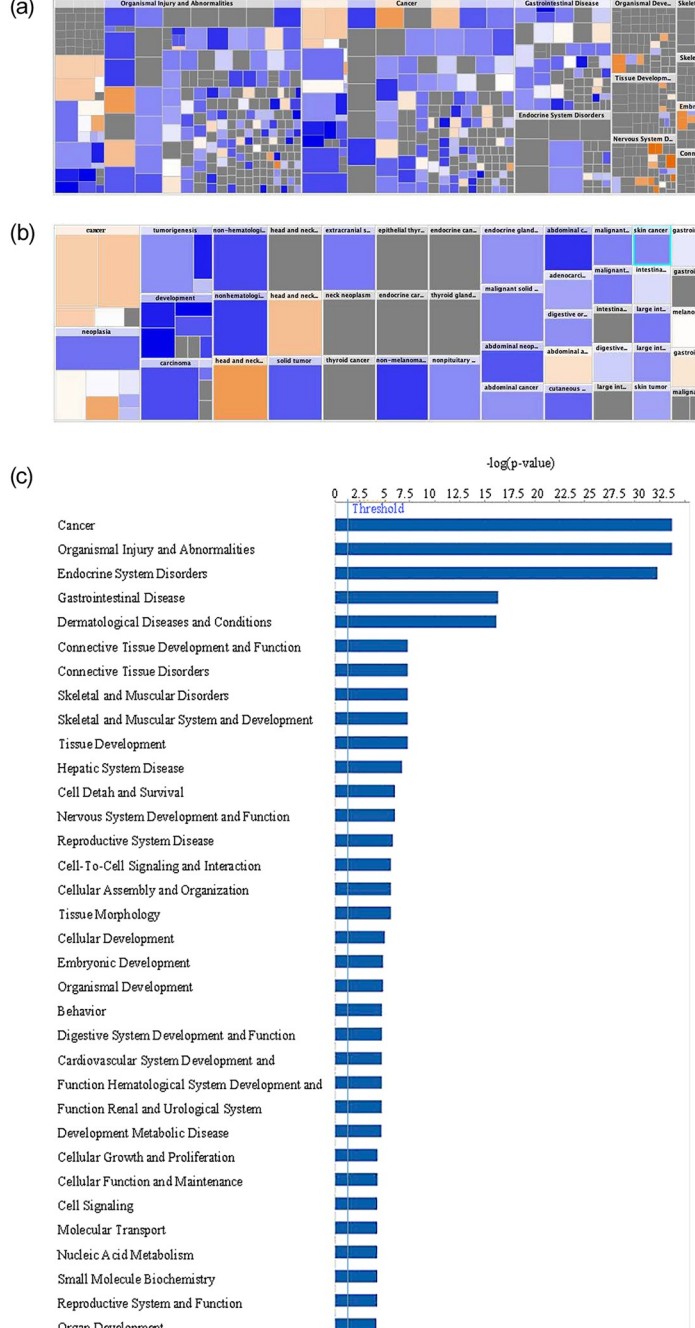

**Fig 5. Gene network analysis between DACH1 mutated and wild-type patients.** (A) A heatmap of the network analysis of genes differentially expressed between *DACH1* mutated patients and wild-type by organ and disease system pathways is shown. The size of the box denotes the -log(p-value). The color of the boxes correlates with the z-score with the intensity of blue representing z ≤ 0 and orange z ≥ 0. Those with the highest z-scores and the greatest p-values include head and neck cancer, head and neck tumor, cancer of secretory structure, and neoplasia of cells. (B) Heatmap of network analysis separated by cancer disease process is shown. This shows an increased z-score in secretory cancers (-log[p-value] = 31.281, z-score = 0.547), head and neck cancers (-log[p-value] = 33.233, z-score = 1.463), abdominal adenocarcinoma (-log[p-value] = 16.306, z-score = 0.328), pelvic cancer (-log[p-value] = 5.439, z-score = 0.391), hyperplasia of the intestinal tract (-log[p-value] = 2.818, z-score = 0.239), prostate cancer (-log [p-value] = 4.883, z-score = 0.291), genital cancer (-log[p-value] = 3.967, z-score = 0.391), and quantity of malignant tumor (-log[p-value] = 2.502, z-score = 1.254). (C) Disease system pathways involved with *DACH1* mutations are shown through network analysis of genes differentially expressed between *DACH1* mutated patients and wild-type.

(a)

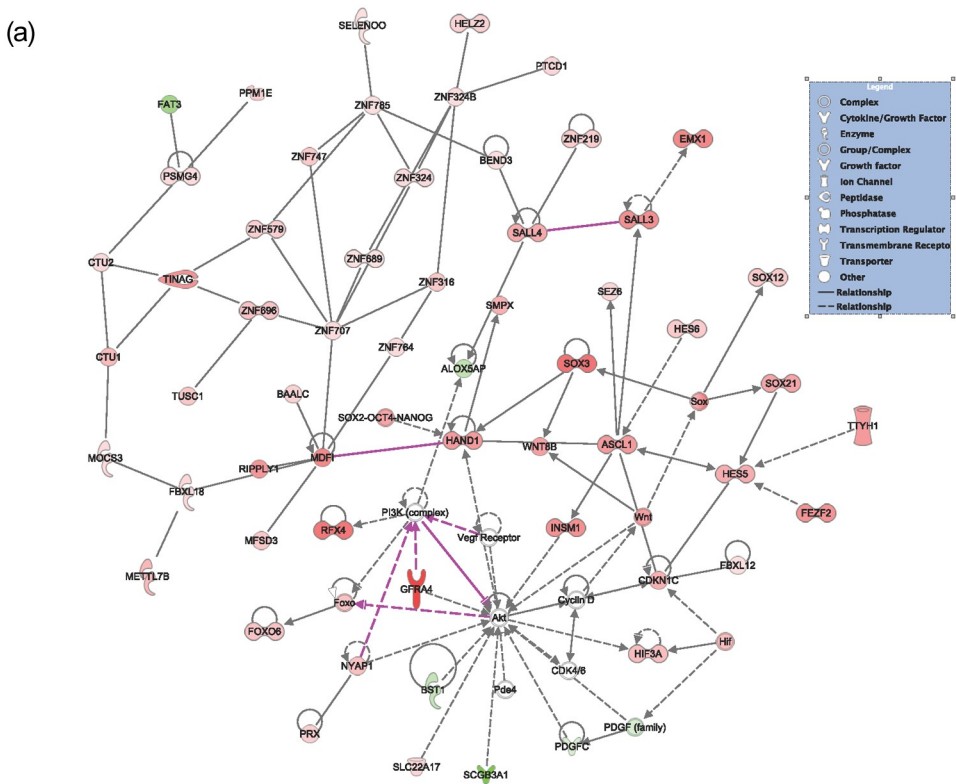

(b)

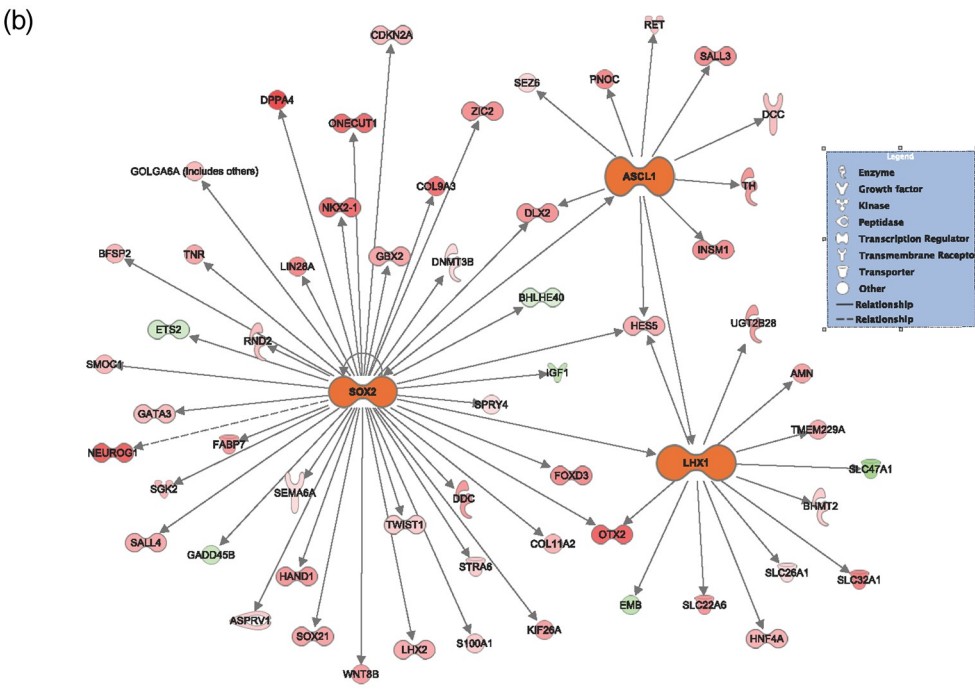

**Fig 6. Network analysis of genes differentially expressed between *DACH1* mutated patients and wild-type.** (A) Network mapping by Qiagen IPA with Global Network Overlay is shown to compare DACH1 mutated patients versus wild-type. (B) Network mapping with statistically significant different gene expression was shown using global network overlay with significance seen in SOX2, ASCL1, and LHX1. Upregulated expression is shown in red, with color intensity corresponding to increased significance. Downregulated expression is notated in green with color intensity again corresponding to increased significance.

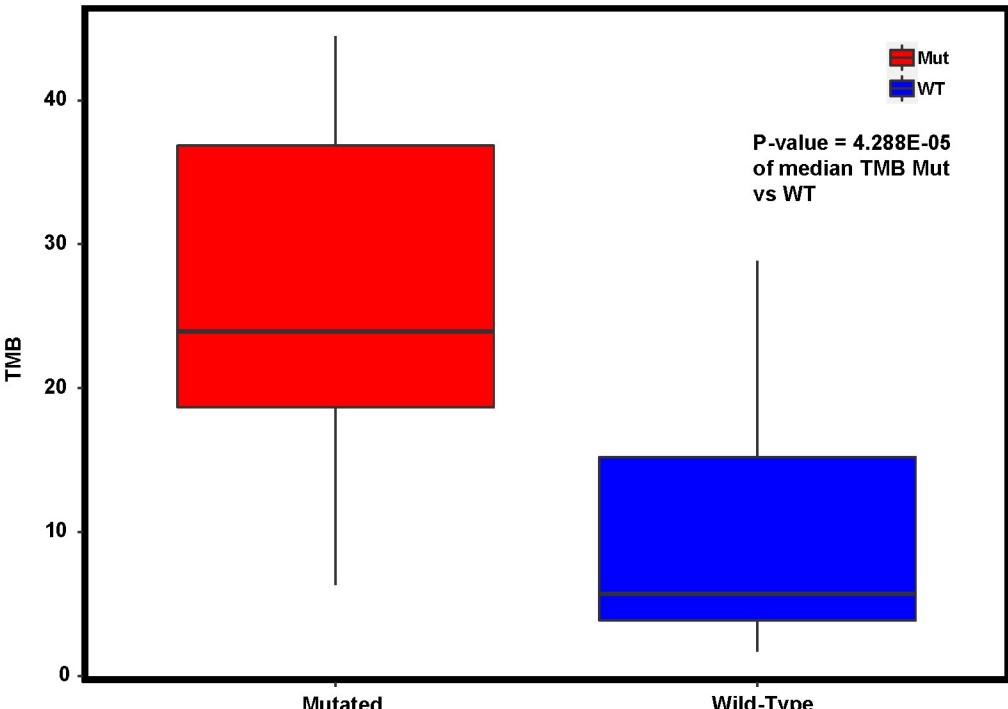

**Fig 7. Tumor mutation burden in *DACH1* mutated patients compared to *DACH1* wild-type as a continuous variable.** The median is statistically different between the two (p-value = 4.288 E-05). TMB high is defined as $\geq$ 20 mutations per megabase.

with endometrial cancer, likely related to the high prevalence of Lynch syndrome associated with colon cancer in the region [25].

TMB $\geq$ 10 received accelerated FDA approval as an indication for treatment with pembrolizumab in malignant solid tumors as studies indicate an improvement in progression-free and overall survival rates with increasing TMB, independent of PD-L1 status [26]. At MCC, *DACH1* mutated patients had a median TMB of 24.0, significantly higher than *DACH1* wildtype patients. In a subgroup analysis of the KEYNOTE-158 trial, patients with TMB $\geq$ 13 had an objective response rate of 37% with an ongoing response $\geq$ 12 months in 58% and $\geq$ 24 months in 50% [27, 28]. With TMB $\geq$ 10, the objective response rate was 29% with the same duration of response. This subgroup included patients both with intact and deficient MMR mechanisms, suggesting TMB alone as an additional indication for treatment with pembrolizumab [28]. Given the median TMB of 24.0 in the *DACH1* population, *DACH1* could serve as a future biomarker for increased TMB and possible treatment indication with checkpoint immunotherapy such as pembrolizumab.

A strength of this investigation is that a significant proportion of our study population has high grade or recurrent disease, which is often a limitation in prior genomic and proteomic analyses of endometrial cancer patients. In addition, the availability of paired RNA Seq and whole exome sequencing allowed for an extensive assessment of differentially altered pathways in those with *DACH1* mutations. Finally, to our knowledge, we are the first to identify the significant co-occurrence of *DACH1* mutations with both *POLE* and Lynch associated genes and an over-representation of these mutations in our Kentucky population. There are also several study limitations to consider. The study sample size is small, with 65 patients total and 12 with

**Table 8. Relationship of gene mutations with microsatellite instability at MCC.**

| Genes | MSI Status | | P-value[a] |
|---|---|---|---|
| | High | Low | |
| **DACH1** | | | 0.1342 |
| Mutant | 3 | 8 | |
| Wild-Type | 4 | 40 | |
| **POLE** | | | 0.001208 |
| Mutant | 7 | 16 | |
| Wild-Type | 0 | 32 | |
| **MLH1** | | | 0.04116 |
| Mutant | 5 | 14 | |
| Wild-Type | 2 | 24 | |
| **MSH2** | | | 1 |
| Mutant | 3 | 19 | |
| Wild-Type | 4 | 29 | |
| **MSH6** | | | 0.1025 |
| Mutant | 5 | 17 | |
| Wild-Type | 2 | 31 | |
| **PMS2** | | | 0.01596 |
| Mutant | 4 | 6 | |
| Wild-Type | 3 | 42 | |

Microsatellite instability in *DACH1* mutated patients compared to wild-type. MSI was calculated by MSISensor2, which assumes MSI-H $\geq$ 20%. Correlation of *DACH1* mutations with microsatellite instability status was not significant (p-value = 0.1342).

[a] P-values calculated using the Fisher's exact test.

*DACH1* mutations, and may be too small to detect possible clinical variables associated with *DACH1* gene mutations. We also compared our cohort to the TCGA PCA, potentially introducing inconsistencies in sequencing and bioinformatics processing. However, given our conservative variant calling and including only known deleterious mutations, we are biasing towards under-calling variants. Clinical characteristics between the TCGA and our population also varied, with more patients with recurrent endometrial cancer in TCGA, making up only 24.6% of the MCC cohort. Nevertheless, the mutation frequency of *PTEN*, *PIK3CA*, and *TP53* was similar between the TCGA and MCC cohorts. In addition, approximately half of included patients had high-grade disease, making them less representative of the uterine cancer population as a whole, but similar to those evaluated by TCGA. Finally, the majority of literature related to *DACH1* in endometrial cancer is at the protein expression level, and the relationship to *DACH1* mutation and protein expression is currently unknown.

## Conclusion

Kentucky has both a high incidence and mortality from endometrial cancer. Compared to the rest of the U.S., Kentucky's population is unique in its genomic and socioeconomic make-up. In part, *DACH1* mutations and enrichments in other co-occurring pathogenic genes may explain these differences. *DACH1* could provide a novel therapeutic target for immunotherapy in this ultrasensitive group of endometrial cancers with increased tumor mutation burden.

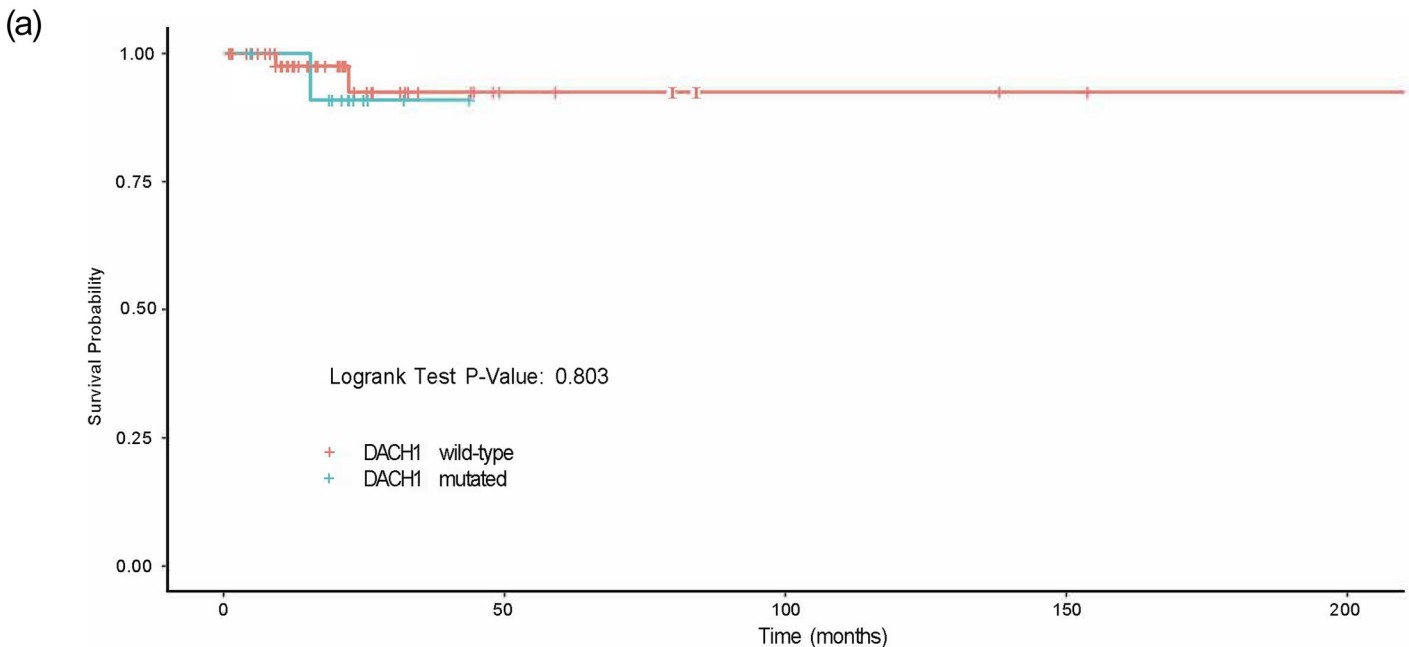

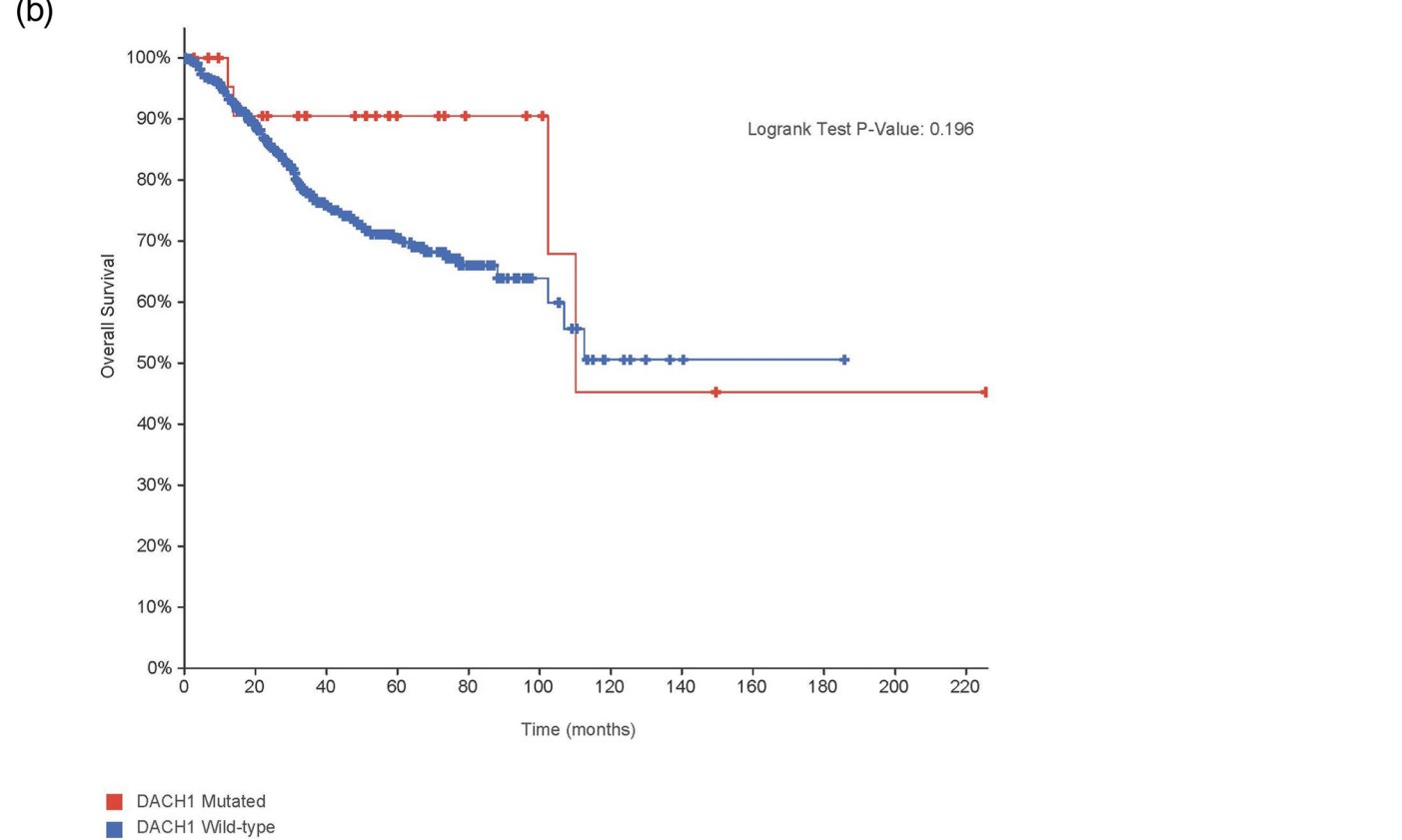

**Fig 8. Overall survival (months) between DACH1 mutated patients versus wild-type.** (A). Overall survival (months) between *DACH1* mutated patients and wild-type at MCC, corrected for stage and grade, were found to be similar with no significant difference (p-value = 0.803) though limited outcome data in *DACH1* mutated patients. (B). Overall survival (months) between *DACH1* mutated patients and wild-type was also evaluated in the TCGA PCA (p-value = 0.196) patients with no significant difference.

## Supporting information

**S1 File. Detailed bioinformatics pipeline document describing the methods used by M2Gen.**
(PDF)

## Author Contributions

**Conceptualization:** McKayla J. Riggs, Mahadev Rao, Jill M. Kolesar.

**Data curation:** McKayla J. Riggs.

**Formal analysis:** McKayla J. Riggs, Nan Lin, Chi Wang, Dava W. Piecoro, Jill M. Kolesar.

**Investigation:** McKayla J. Riggs, Nan Lin, Jill M. Kolesar.

**Methodology:** McKayla J. Riggs, Chi Wang, Dava W. Piecoro, Oliver A. Hampton, Jill M. Kolesar.

**Project administration:** Jill M. Kolesar.

**Software:** Nan Lin, Oliver A. Hampton.

**Supervision:** Rachel W. Miller, Frederick R. Ueland, Jill M. Kolesar.

**Writing – original draft:** McKayla J. Riggs, Nan Lin, Jill M. Kolesar.

**Writing – review & editing:** McKayla J. Riggs, Chi Wang, Dava W. Piecoro, Rachel W. Miller, Oliver A. Hampton, Mahadev Rao, Frederick R. Ueland, Jill M. Kolesar.

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
