## [Decision Letter · Decision Letter 0]

22 Oct 2020

PONE-D-20-31424

*DACH1* mutation frequency in endometrial cancer is associated with high tumor mutation burden

PLOS ONE

Dear Dr. Kolesar,

Thank you for submitting your manuscript to PLOS ONE. After careful consideration, we feel that it has merit but does not fully meet PLOS ONE’s publication criteria as it currently stands. Therefore, we invite you to submit a revised version of the manuscript that addresses the points raised during the review process.

We look forward to receiving your revised manuscript.

Kind regards,

Sumitra Deb, PhD

Academic Editor

PLOS ONE

Journal Requirements:

2. We note that Figure 1 in your submission contain copyrighted images. All PLOS content is published under the Creative Commons Attribution License (CC BY 4.0), which means that the manuscript, images, and Supporting Information files will be freely available online, and any third party is permitted to access, download, copy, distribute, and use these materials in any way, even commercially, with proper attribution. For more information, see our copyright guidelines: http://journals.plos.org/plosone/s/licenses-and-copyright.

2.1.         You may seek permission from the original copyright holder of Figure 1 to publish the content specifically under the CC BY 4.0 license.

2.2.    If you are unable to obtain permission from the original copyright holder to publish these figures under the CC BY 4.0 license or if the copyright holder’s requirements are incompatible with the CC BY 4.0 license, please either i) remove the figure or ii) supply a replacement figure that complies with the CC BY 4.0 license. Please check copyright information on all replacement figures and update the figure caption with source information. If applicable, please specify in the figure caption text when a figure is similar but not identical to the original image and is therefore for illustrative purposes only.

3. In your Methods section, please provide additional information about the participant recruitment method and the demographic details of your participants. Please ensure you have provided sufficient details to replicate the analyses such as: a) the recruitment date range (month and year), b) a description of any inclusion/exclusion criteria that were applied to participant recruitment, c) a description of how participants were recruited.

4. In the ethics statement in the manuscript and in the online submission form, please provide additional information about the patient records used in your study, including: a) whether all data were fully anonymized before you accessed them; b) the date range (month and year) during which patients' medical records were accessed.

5. Please provide the accession number and/or URL for any datasets obtained from the TCGA database.

6. Thank you for stating the following in the Competing Interests section:

"I have read the journal's policy and the authors of this manuscript have the following competing interests: Dr. Piecoro’s spouse is employed by Exelixis, Inc. Dr. Hampton, is employed by M2Gen, a for profit company focused on providing oncology health informatics solutions to accelerate cancer treatment discovery, development, and delivery by leveraging clinical and molecular data."

We note that one or more of the authors are employed by a commercial company: M2Gen, Exelixis, Inc.

6.1. Please provide an amended Funding Statement declaring this commercial affiliation, as well as a statement regarding the Role of Funders in your study. If the funding organization did not play a role in the study design, data collection and analysis, decision to publish, or preparation of the manuscript and only provided financial support in the form of authors' salaries and/or research materials, please review your statements relating to the author contributions, and ensure you have specifically and accurately indicated the role(s) that these authors had in your study. You can update author roles in the Author Contributions section of the online submission form.

6.2. Please also provide an updated Competing Interests Statement declaring this commercial affiliation along with any other relevant declarations relating to employment, consultancy, patents, products in development, or marketed products, etc.  

7. In your Data Availability statement, you have not specified where the minimal data set underlying the results described in your manuscript can be found. PLOS defines a study's minimal data set as the underlying data used to reach the conclusions drawn in the manuscript and any additional data required to replicate the reported study findings in their entirety. All PLOS journals require that the minimal data set be made fully available. For more information about our data policy, please see http://journals.plos.org/plosone/s/data-availability.

8. Please upload a copy of Supporting Information Appendix 1 which you refer to in your text on page 8 and 37.

Reviewers' comments:

Reviewer's Responses to Questions

**Comments to the Author**

1. Is the manuscript technically sound, and do the data support the conclusions?

Reviewer #1: Partly

Reviewer #2: Yes

2. Has the statistical analysis been performed appropriately and rigorously? 

Reviewer #1: Yes

Reviewer #2: Yes

3. Have the authors made all data underlying the findings in their manuscript fully available?

Reviewer #1: No

Reviewer #2: Yes

4. Is the manuscript presented in an intelligible fashion and written in standard English?

Reviewer #1: Yes

Reviewer #2: Yes

5. Review Comments to the Author

Reviewer #1: The title of this paper is DACH1 mutation frequency in endometrial cancer is associated with high tumor mutation burden. Based on that title a couple required elements are missing and pieces are out of place. Endometrial cancer, immediately thinking subsets (see major point 4). These subsets are somewhat addressed in table 6 (and are a confounder of table 3) but need to be flushed out in more detail and much earlier in the manuscript. The TCGA has MSI information in the unrestricted access clinical tables so this comparison is doable. It's possible DACH1 doesn't fit into neatly into those subsets and that could justify all the pathway analysis but right now I wonder why the pathway analysis is there. I'd like to see a much deeper dive into analyses that support the title of the paper (e.g. the mutations in DACH1 or TMB by DACH1 status and also analyzed within subgroups). What's reported is technically correct but it lacks a greater context.

Major points:

1) Data availability, the authors should state what data is available in addition to the current statements about what data is not available.

2) The bioinformatics method section only describes in very general terms the RNAseq methodology. Is Figure 1 copied from a brochure? Ideally would describe with references and version numbers. False discovery rate method for Figure 3 should be described here as well. Considering most of the figures are bioinformatics analyses, this section needs to be strengthened significantly.

3) Table 2 is screaming for comparison of MSS to MSI-H, ah there it is in Table 6.

4) In comparing to TCGA there needs to be some description or normalization of cohorts. What fraction of each cohort is POLE, MSI-H, and other (copy number high vs low can be a judgement call so lets ignore it). Then what fraction of POLE, MSI-H and other is DACH1 mutated. This flaw permeates the manuscript and especially figure 8 since the TMB covariate is not controlled for.

5) There are few enough DACH1 mutations that they should all be listed with protein effects. Are these nonsense alterations? Splicing? Silent? Are the amino acid changes consistent with loss of function (e.g. charge switch)?

Minor points:

1) Line 194: p = 0.053 is approaching significance. Saying it is marginally significant makes me think the p value is 0.04999.

2) Figure 2 would look better as a lollipop plot. If the data is formatted properly, cBioportal will make one for you.

3) Figure 3 would benefit from labeling some of the genes called out in the text.

4) Figures 4, 5, 6 have copyright QIAGEN all rights reserved on them. Are the authors allowed to publish these plots in PLOS ONE?

Reviewer #2: The authors have done a thorough job of doing all the relevant data analysis and the manuscript has also been well written. I have no additional comments on the paper and I recommend that the manuscript be accepted.

6. PLOS authors have the option to publish the peer review history of their article (what does this mean?). If published, this will include your full peer review and any attached files.

Reviewer #1: No

Reviewer #2: No

---

## [Author Response · Author response to Decision Letter 0]

8 Dec 2020

Reviewer #1: The title of this paper is DACH1 mutation frequency in endometrial cancer is associated with high tumor mutation burden. Based on that title a couple required elements are missing and pieces are out of place. Endometrial cancer, immediately thinking subsets (see major point 4). These subsets are somewhat addressed in table 6 (and are a confounder of table 3) but need to be flushed out in more detail and much earlier in the manuscript. The TCGA has MSI information in the unrestricted access clinical tables so this comparison is doable. It's possible DACH1 doesn't fit into neatly into those subsets and that could justify all the pathway analysis but right now I wonder why the pathway analysis is there. I'd like to see a much deeper dive into analyses that support the title of the paper (e.g. the mutations in DACH1 or TMB by DACH1 status and also analyzed within subgroups). What's reported is technically correct but it lacks a greater context.

This was included as requested. 

Major points:

1) Data availability, the authors should state what data is available in addition to the current statements about what data is not available.

This has been addressed above. Data must be requested through the Kentucky Cancer Registry as described as they are the third party honest broker due to the sensitivity and legal implications of using such data. 

2) The bioinformatics method section only describes in very general terms the RNAseq methodology. Is Figure 1 copied from a brochure? Ideally would describe with references and version numbers. False discovery rate method for Figure 3 should be described here as well. Considering most of the figures are bioinformatics analyses, this section needs to be strengthened significantly.

Figure 1 is not copied from a brochure but is provided by M2Gen to depict their bioinformatics pipeline with additional details described in supplemental appendix 1. 

3) Table 2 is screaming for comparison of MSS to MSI-H, ah there it is in Table 6.

This has been added as requested.

4) In comparing to TCGA there needs to be some description or normalization of cohorts. What fraction of each cohort is POLE, MSI-H, and other (copy number high vs low can be a judgement call so lets ignore it). Then what fraction of POLE, MSI-H and other is DACH1 mutated. This flaw permeates the manuscript and especially figure 8 since the TMB covariate is not controlled for.

Table 2b was added to address these concerns. Table 3 was updated to reflect and allow for normalization of the cohorts. 

5) There are few enough DACH1 mutations that they should all be listed with protein effects. Are these nonsense alterations? Splicing? Silent? Are the amino acid changes consistent with loss of function (e.g. charge switch)?

This has been added as an additional table. 

Minor points:

1) Line 194: p = 0.053 is approaching significance. Saying it is marginally significant makes me think the p value is 0.04999. 

This is reworded to further clarify. 

2) Figure 2 would look better as a lollipop plot. If the data is formatted properly, cBioportal will make one for you.

This has been changed to a lollipop plot as requested.

3) Figure 3 would benefit from labeling some of the genes called out in the text.

This has been updated to include the genes called out in the text as requested. 

4) Figures 4, 5, 6 have copyright QIAGEN all rights reserved on them. Are the authors allowed to publish these plots in PLOS ONE?

The QIAGEN software was published by our department, allowing us the rights to publish the plots of our data using their software. 

Reviewer #2: The authors have done a thorough job of doing all the relevant data analysis and the manuscript has also been well written. I have no additional comments on the paper and I recommend that the manuscript be accepted.

No additional requests noted from Reviewer #2. 

6. PLOS authors have the option to publish the peer review history of their article (what does this mean?). If published, this will include your full peer review and any attached files.

Do you want your identity to be public for this peer review? For information about this choice, including consent withdrawal, please see our Privacy Policy.

Reviewer #1: No

Reviewer #2: No

---

## [Decision Letter · Decision Letter 1]

14 Dec 2020

*DACH1* mutation frequency in endometrial cancer is associated with high tumor mutation burden

PONE-D-20-31424R1

Dear Dr. Kolesar,

We’re pleased to inform you that your manuscript has been judged scientifically suitable for publication and will be formally accepted for publication once it meets all outstanding technical requirements.

Kind regards,

Sumitra Deb, PhD

Academic Editor

PLOS ONE

Additional Editor Comments (optional):

Reviewers' comments:

Reviewer's Responses to Questions

**Comments to the Author**

1. If the authors have adequately addressed your comments raised in a previous round of review and you feel that this manuscript is now acceptable for publication, you may indicate that here to bypass the “Comments to the Author” section, enter your conflict of interest statement in the “Confidential to Editor” section, and submit your "Accept" recommendation.

Reviewer #1: (No Response)

2. Is the manuscript technically sound, and do the data support the conclusions?

Reviewer #1: Yes

3. Has the statistical analysis been performed appropriately and rigorously? 

Reviewer #1: Yes

4. Have the authors made all data underlying the findings in their manuscript fully available?

Reviewer #1: Yes

5. Is the manuscript presented in an intelligible fashion and written in standard English?

Reviewer #1: Yes

6. Review Comments to the Author

Reviewer #1: All comments have been addressed and the data sharing is a satisfactory balance between access and patient privacy.

One minor comment: Line 286, Hochberg is spelled with an 'h'

7. PLOS authors have the option to publish the peer review history of their article (what does this mean?). If published, this will include your full peer review and any attached files.

Reviewer #1: No

---

## [Editor Report · Acceptance letter]

17 Dec 2020

PONE-D-20-31424R1 

*DACH1* mutation frequency in endometrial cancer is associated with high tumor mutation burden 

Dear Dr. Kolesar:

I'm pleased to inform you that your manuscript has been deemed suitable for publication in PLOS ONE. Congratulations! Your manuscript is now with our production department. 

Kind regards, 

on behalf of

Dr. Sumitra Deb 

Academic Editor

PLOS ONE